# Recent Advances in Catalysts and Membranes for MCH Dehydrogenation: A Mini Review

**DOI:** 10.3390/membranes11120955

**Published:** 2021-12-01

**Authors:** Durga Acharya, Derrick Ng, Zongli Xie

**Affiliations:** CSIRO Manufacturing, Private Bag 10, Clayton South, Melbourne, VIC 3169, Australia; durga.acharya@csiro.au (D.A.); derrick.ng@csiro.au (D.N.)

**Keywords:** methylcyclohexane, dehydrogenation, catalytic membrane reactor, catalyst, membrane, liquid organic hydrogen carrier (LOHC)

## Abstract

Methylcyclohexane (MCH), one of the liquid organic hydrogen carriers (LOHCs), offers a convenient way to store, transport, and supply hydrogen. Some features of MCH such as its liquid state at ambient temperature and pressure, large hydrogen storage capacity, its well-known catalytic endothermic dehydrogenation reaction and ease at which its dehydrogenated counterpart (toluene) can be hydrogenated back to MCH and make it one of the serious contenders for the development of hydrogen storage and transportation system of the future. In addition to advances on catalysts for MCH dehydrogenation and inorganic membrane for selective and efficient separation of hydrogen, there are increasing research interests on catalytic membrane reactors (CMR) that combine a catalyst and hydrogen separation membrane together in a compact system for improved efficiency because of the shift of the equilibrium dehydrogenation reaction forwarded by the continuous removal of hydrogen from the reaction mixture. Development of efficient CMRs can serve as an important step toward commercially viable hydrogen production systems. The recently demonstrated commercial MCH-TOL based hydrogen storage plant, international transportation network and compact hydrogen producing plants by Chiyoda and some other companies serves as initial successful steps toward the development of full-fledged operation of manufacturing, transportation and storage of zero carbon emission hydrogen in the future. There have been initiatives by industries in the development of compact on-board dehydrogenation plants to fuel hydrogen-powered locomotives. This review mainly focuses on recent advances in different technical aspects of catalytic dehydrogenation of MCH and some significant achievements in the commercial development of MCH-TOL based hydrogen storage, transportation and supply systems, along with the challenges and future prospects.

## 1. Introduction

Hydrogen-based energy systems such as fuel cells have drawn considerable attention due to their high energy efficiency and absence of polluting emissions, making them a promising candidate for power sources [1,2] in different applications, including public and personal transportations. In recent years, several leading automobile manufacturers have released vehicles models powered by fuel cell. One of the major challenges in integrating fuel cells into the energy system is the overcoming the inconveniences of hydrogen storage as highly compressed gas in large and thick tanks or the cost of cryogenic liquid hydrogen, especially for portable and on-board uses. Other approaches for hydrogen storage such as its absorption in high surface carbon-based materials such as nanotubes, nanorods, graphite and activated carbon [3,4], and other non-carbon materials such as mesoporous silica (MCM-41) [5], high entropy alloys [6,7], metallic organic frameworks (MOFs) [8,9,10] have emerged, but the main drawback is that their gravimetric hydrogen capacity is lower than 1% at ambient temperature and high pressure up to 100 bar, which is far below the respective gravimetric and volumetric performance target of 4.5 wt% and 30 g/L and the ultimate target of 6.5 wt% and 50 g/L at 233–358 K and 5–12 bars set by the US Department of Energy (DoE) for usable hydrogen storage capacity of on-board or mobile storage systems for the year 2020. Some of the MOFs achieve gravimetric hydrogen capacity on par to or even better than the DoE’s target at much higher pressures and at cryogenic temperature (77 K) but as the temperature is increased, to as high as 160 K, the capacity sharply drops. Together with some other metal hydrides (MH) magnesium hydride (MgH_2_) is considered as one of the potential hydrogen storage materials because of excellent hydrogen storage capacity (7.6 wt%), low price, rich resource metals, and excellent reversibility, but some unfavorable properties such as high thermodynamic stability of the hydride, slow dehydrogenation, and high temperature (300–400 °C) required for both dehydrogenation and hydrogenation make them unfeasible for practical use. Different approaches have been taken into consideration to address these challenges, such as by using hydride of metal alloys and using nanostructured metal hydrides of high porosity [11,12,13,14].

Chemical storage of hydrogen in the form of hydrogen-rich cyclic or heterocyclic liquid organic compounds known as liquid organic hydrogen carrier (LOHC) is emerging as a viable option [15,16,17,18,19,20,21,22,23,24]. Hydrogen-rich LOHC are typically a cyclic or heterocyclic organic compound that can undergo dehydrogenation to produce hydrogen and an unsaturated or aromatic compound. LOHCs, because of their favorable thermodynamic and kinetics of hydrogen storage/release, larger storage capacity (6–8 wt%) compared to metal hydrides (<3%), and the possibility of recycling dehydrogenated products into hydrogenated form for repeated use, have drawn considerable attention for storage and transport of hydrogen. Moreover, because of LOHC’s physical properties similar to gasoline or diesel, existing transportation infrastructure developed for gasolines can be used for LOHC with very little changes (Figure 1).

Lower endothermicity of heterocyclic organic liquids compounds facilitates the dehydrogenation process, thus making these heterocycles potentially viable as a hydrogen storage substrate but their degradation by C–N cleavage, disproportionation, alkyl transfer, and other side reactions that occur during the dehydrogenation and hydrogenation affect their dehydrogenation and hydrogenation efficiency [20,25,26,27]. Because of this reason, cyclic hydrocarbon based LOHCs have drawn considerable attention.

Among the most investigated homocyclic LOHCs, namely, cyclohexane (CH), methylcyclohexane (MCH), and decalin (DEC), MCH is the most promising due to its high affinity with existing conventional transport, storage, and distribution system in spite of its lower hydrogen storage capacity (6.2 wt% or 47.3 kgH_2_ m^−3^) compared to that of CH (7.2 wt%) and DEC (7.3 wt%) because of the carcinogenic toxicity of benzene (BEN, product from CH) and solid phase of naphthalene (NAP, product from DEC) at ambient conditions that poses extra inconveniences in storage and handling [24]. Very low freezing temperature of MCH (−126.6 °C) and its dehydrogenated form toluene (TOL) (−95 °C) than that of CH (6.5 °C) and benzene (5.5 °C) add further convenience in storage and handling of MCH and TOL—even at subzero temperature conditions. On the other hand, high boiling temperatures of MCH and TOL (>100 °C) make them less volatile and therefore less hazardous to handle at normal temperature and pressure as suggested by a recent thermal hazard study [28]. However, the high aquatic toxicity and the poor biodegradability of MCH and the reproductive toxicity of TOL suggests that the LOHC system could generally present a high hazard to humans as well as the environment, and make them inferior to current diesel oil based energy system [29]. Table 1 shows some properties of MCH, CH, and DEC and their respective hydrogen-lean counterparts.

Higher equilibrium conversion of MCH than of CH also means that MCH dehydrogenation produces more hydrogen at a given temperature or that stripping of hydrogen can be carried out at a lower temperature to achieve the same level of conversion. There are already some commercialization activities currently being advanced for this emerging LOHC system. For example, Chiyoda Corporation successfully demonstrated the world’s first international hydrogen supply chain system based on MCH-TOL system in 2020 by transporting and storing over 100 tons of hydrogen over 10 months [31].

Shifting equilibrium dehydrogenation reaction forward by efficient and continuous removal of hydrogen from the reaction mixtures can maximize the production of hydrogen [32,33], which could be achieved in catalytic membrane reactor (CMR) systems that combine the catalytic reactor for dehydrogenation and a separating mechanism that selectively removes hydrogen from the system. In recent years, there have been increasing research interests on catalytic dehydrogenation MCH using various catalysts in specially designed CMR that incorporates porous inorganic membranes for the continuous removal hydrogen. CMR systems operating at high temperatures (above 300 °C) not only produce very high purity hydrogen without requiring another hydrogen separation mechanism, but also operate with better energy efficiency requiring lesser amount of energy. Byun et al. recently reported that CMR is more cost effective than packed bed reactor, and that incorporation of membranes with high permeation significantly reduced the overall cost [34].

Some review articles published recently cover specific aspects of MCH dehydrogenation process, such as catalysts used for the dehydrogenation of MCH [35] or LOHC in general [36], specific type of membranes for hydrogen purification [37] or extensively covering nearly all types of membranes for hydrogen purification, with some of them not relevant to the conditions of MCH dehydrogenation [38]. This review mainly focuses on recent advances in different aspects of catalytic dehydrogenation of MCH, including catalysts used for MCH dehydrogenation reaction, inorganic porous membranes for separation of hydrogen from reaction mixture and design of CMR for MCH-TOL system. Moreover, some significant achievements have been made recently in the commercial development of MCH-TOL based hydrogen storage, transportation, and supply systems, with a growing number of industries being involved in the development of LOHC based hydrogen systems. These developments initiatives along with the challenges and future prospects have also been covered.

## 2. Catalysts for MCH Dehydrogenation

Dehydrogenation of MCH to TOL and hydrogen is particularly useful for a hydrogen storage system [39]. The reaction is reversible, highly endothermic as shown in Equation (1), and when the reaction is conducted under high temperatures, by-products other than toluene and hydrogen are likely to form. Therefore, the reaction requires a suitable catalyst and condition to achieve favorable MCH conversion with high selectivity towards TOL and hydrogen.
C_7_H_14_ → C_7_H_8_ + 3H_2_, ΔH^0^_298_ = 204.8 kJ mol^−1^(1)

From the late 1960s to 2000s, researchers primarily exploited supported Pt-containing catalysts for MCH dehydrogenation reaction. For instance, in 1966, Ritchie used platinum on alumina, catalyzing the MCH dehydrogenation process, whilst in 1977 Wolf studied the catalyst poisoning effect of a similar reaction [40,41]. Screening through the literature, platinum evolves as the crowned champion as the catalyst in the MCH dehydrogenation, and it has been utilized in a small quantity, typically less than 3 wt%, either in a mono or multi-metallic components catalysts [40,42,43,44,45,46,47,48]. Thenceforth, other noble and non-noble metal catalysts have been progressively utilized in conjunction with a variety of supports or carriers to enhance the dispersion of active catalysts within the body.

Other reliable noble metals which have proven their excellent dehydrogenation are palladium and iridium, which acquire similar atomic spacing to platinum. Horikoshi et al. reported that by using a palladium catalyst supported on activated carbon, which was assisted by microwave heating, MCH dehydrogenation conversion of 95–99% was achieved at optimal flow rates around 2 min reaction time as opposed to 35 min for conventional heating [49]. Cromwell et al. compared the activity of palladium with iridium, platinum, and nickel, all supported on ultrastable Y (USY) zeolite molecular sieves. They reported that the iridium catalyst was more effective for MCH dehydrogenation than the classical platinum based catalyst (both having similar metal loadings and surface properties) owing to its increased hydrogenolysis “kink” sites [50]. Overall, palladium or iridium are however, to a lesser extent, being utilized as active catalysts on their own compared to platinum. Researchers were more inclined in incorporating them into porous catalytic membrane reactor system [44].

Recently, non-noble metals such as nickel, molybdenum, zinc, cerium, and copper have also been emerging as promising candidates for MCH dehydrogenation. For instance, Hatim et al. compared MCH dehydrogenation using 50 wt % nickel on alumina versus 0.5 wt% platinum on alumina and concluded that for tests conducted in a fixed bed reactor, platinum is most efficient at lower temperatures below 300 °C but nickel yielded higher MCH conversions at higher temperatures over 500 °C. They also impregnated the catalyst onto porous alumina hollow fiber for MCH dehydrogenation, and concluded that asymmetric alumina hollow fibers significantly increase the efficiency of the reaction over conventional fixed bed reactor [44]. Using a sol-gel method, Boufaden et al. have managed to synthesize partially reduced molybdenum-silica catalyst. Their best performing catalyst (10 mol.% molybdenum in respect to silica) had the highest activity and 90% selectivity towards toluene, attributed to its great balance of existence of MoO_2_ and MoO_3_ as well as lesser coke formation on the catalyst surface [51]. In other cases, non-noble metals were used as additives to enhance the activity of noble catalyst. Active noble metal such as platinum is commonly added with a less active or inactive second metal to inhibit undesired side reactions such as C–C cleavage for selectivity enhancement in alkane dehydrogenation, where the catalytic activity could be compromised [52]. Mori et al. investigated the effects of adding zinc, tin, cerium, gallium, and manganese to platinum catalyst. Results showed that zinc was the best additive for platinum-alumina catalyst in increasing MCH conversion and decreasing selectivity for by-products. They claimed that the enhancement was due to the increase of electron-donating ability when zinc was added [47]. On top of that, the effect of various supports (alumina, titania, silica, and activated carbon) was also systematically studied, and it was concluded that alumina and activated carbon were both performing most satisfactorily in terms of MCH conversion and toluene selectivity. However, when zinc was added, platinum/activated carbon catalyst formed aggregated PtZn alloy species, which ultimately compromised the catalytic activity [47]. Other recently emerging but similar ideas were presented, where researchers added a second or third metal to platinum. For example, Nakaya et al. employed the addition of iron and zinc [52], Zhang et al. added copper [53], while Sebastian et al. added gallium but the reaction was conducted under liquid alloy condition supported on silica [54] which is a new concept. The testing conditions of the three catalysts were in a similar vicinity of around 400–450 °C at 1 bar. Iron-zinc/platinum and copper/platinum were performed at a higher level with >90% MCH conversion while gallium/platinum only attained 16.5%. The use of second or third metal alloyed with platinum is said to enhance the decoking capability/coke resistance [53], and at the same time assist in toluene desorption by both ligand and ensemble effects [52]. Table 2 lists the performance of the reported catalysts employed for MCH dehydrogenation in recent years.

## 3. Membranes for MCH Dehydrogenation

The MCH dehydrogenation reaction for hydrogen production is a thermodynamic equilibrium-limited endothermic reaction (Equation (1)). It is normally conducted at reaction temperatures higher than 350 °C due to its kinetic and thermodynamic limitations. The equilibrium can be shifted to promote hydrogen production by extraction of the product, which can be achieved by using a highly selective membrane to selectively remove hydrogen or TOL from the product, shifting the MCH dehydrogenation reaction towards the product side and producing pure hydrogen simultaneously. Wide research and the advance in hydrogen separation membranes has made separating hydrogen from the product a more viable approach for MCH dehydrogenation. Polymeric (organic) membranes are very common and cost-effective, however, their relatively low operating temperature range (commonly below 100 °C with some polymers below 200 °C) makes them unsuitable as hydrogen separation membrane at high temperature environment required for MCH dehydrogenation. Therefore, only membranes made of inorganic materials that can withstand high temperatures for practical applications are included in this review. In general, there are two parameters that determine the performance of a hydrogen selective membrane: (1) hydrogen permeance and (2) separation factors or selectivity of hydrogen compared with other gases that are to be separated. These factors are closely related to membrane properties. Figure 2 shows the major/common types of inorganic membranes used in hydrogen separation.

Dense metallic membrane such as palladium or its alloys [37,61,62] allow for dissociation of hydrogen at the surface and then allow the atomic hydrogen to diffuse to the other side, where re-association to molecular hydrogen occurs. A dense structure prevents the passage of large atoms and molecules, which means that these membranes provide excellent selectivity for hydrogen that translates into production of high purity hydrogen. However, the high cost of metal itself and brittleness arising from repeated use of Pd at high temperature are major disadvantages. Other metals that offer hydrogen diffusivity are tantalum, niobium, and vanadium, which are comparatively cheap. Dense ceramic oxides materials with high proton and electronic conductivity (mostly perovskite-type oxides but also pyrochlorates, niobates, tantalates, and tungstates with high proton and electronic conductivities) are also capable of providing high purity hydrogen separation, but they are still limited to lab scale use and in only few applications. Recent review papers reported advances in the fabrication of various types of membranes for the separation of hydrogen and other gases [37,38,63,64,65,66,67,68].

Porous inorganic membranes that provide hydrogen separation by “molecular sieve mechanism” have high thermal, mechanical, chemical stabilities, and controllable pore size, and are therefore a natural choice for the separation of hydrogen at the high temperature environment at which dehydrogenation reactors work. The ease of fabrication of these porous inorganic membranes from low-cost materials such as silica and the possibility to integrate the dehydrogenation catalyst into the porous membranes at the fabrication stage to make them a compact ‘bimodal’ system is another advantage. Such membrane reactors with hydrogen-selective membranes shift the equilibrium reaction toward higher conversion by extracting the produced hydrogen from the reaction side to the permeation side, thus producing high-purity hydrogen in one step, without any requirement for post-treatment.

There are three types of porous membranes, namely, zeolite, carbon sieve, and silica-based membranes, being reported for hydrogen separation (Figure 2). Crystalline inorganic framework structures of zeolite that produce a molecular sieve with uniform, molecular-sized pores serve as an excellent gas separation membrane, beside their common application as bulk catalysts and adsorbents. When the zeolite pore size distribution falls between the molecular sizes of the gases in the feed side, gas separation occurs by size exclusion mechanism. Gas separation is achieved by the combination of three processes: adsorption of gas molecules on the zeolite surface and then diffusion through the membrane to the surface of the other side where desorption of gas molecules occurs. Excellent thermal as well as mechanical and chemical stabilities, and the ability to be regenerated without loss to performance make zeolite the membrane of superior performance. However, the high cost of membrane production and the presence of intercrystal pores with sizes larger than the zeolitic pores, formed inherently in polycrystalline zeolite membrane, are major drawbacks in the application of zeolite membrane is industrial level [38,63].

Carbon sieve membranes are prepared by high temperature treatment of organic polymer such as polyimide either in a supported or unsupported form in an inert atmosphere (carbonization) to obtain the carbon sieve, which serves as a separation membrane. Unlike other membranes used for hydrogen separations, carbon sieve membranes purify hydrogen by a rejection mechanism arising from low critical temperature and a small kinetic diameter of hydrogen, thus building up hydrogen concentration in the retentate side while allowing the contaminant to permeate through the membrane. Carbon sieve membranes are suitable in light gas separation, however their brittleness and very high cost of production that arises from the use of expensive polyimide as a raw material limit their use in industrial applications. Moreover, these membranes exhibit complications related to their performance stability towards oxygen and even more towards humidity [38,63].

Silica membranes have been commonly used for hydrogen separation because of the low cost of production, ease of fabrication, scalability, and stability. Silica-based porous membranes can be fabricated via chemical vapor deposition (CVD) and sol-gel route. CVD involves the deposition of silica in the form of a thin film from a vapor of precursor such as tetraethyl orthosilicate (TEOS) sprayed on a heated substrate surface [69,70]. The substrate usually consists of an alumina (α- or γ-alumina) or a porous glass such as Vycor glass, having a pore diameter of approximately 4 nm. If alumina has a large pore diameter such as 0.1 micron, the substrate is coated with a layer of γ-alumina to bring the pore size to around 4 nm before carrying out CVD to form a final layer [71]. To make the membrane hydrogen selective, the pore size of the top silica membrane should be around 0.5 nm. There are two methods of CVD for supplying and depositing precursors on the substrate: single-side and counter-diffusion. In single-side deposition, the precursor is supplied from one side on the substrate surface to form a layer, while vacuuming the other side to improve the deposition and form a pinhole-free membrane. In counter-diffusion CVD, two kinds of reactants are supplied from the opposite side of a substrate and pore sizes and effective membrane thickness are controlled by changing the reactants and reaction conditions [69,72,73].

The mostly used sol-gel route involves a series of hydrolysis of alkoxysilane dissolved in their parent alcohols in the presence of a calculated amount of water and a mineral acid or a base as catalyst to produce silanol groups, which undergo condensation reactions to form silica as the final product [74]. The use of acid catalysts produces a three-dimensional network called gel, whereas in presence of base as the catalyst, colloidal microspheres are formed. In membranes formed by the polymeric route, pore sizes are considered to be made up of spaces within amorphous silica networks, whereas in membranes formed by colloidal sol-gel route spaces between colloidal particles are assumed to be pores. The polymeric route for the control of pore sizes is preferable in the preparation of small gas-separation membranes, especially for hydrogen-separation membranes [75]. TEOS has been the most commonly used precursor in the synthesis of silica sol [76,77], but in recent years, many other precursors such as bis(triethoxysilyl)ethane (BTESE) and its homologous members [78,79,80,81,82,83,84,85], 1,1,3,3-tetraethoxy-1,3-dimethyldisiloxane (TEDMDS) [86], triethoxysilane (TRIES) [87] have drawn considerable attention for better stability and improved hydrogen permeability. Compared to silica membranes prepared from TEOS, organosilica membranes derived from bridged bis-silyl precursors with (Si–CH_2_–Si) or Si–(CH_2_)_2_–Si) as bridging units such bis(triethoxysilyl)methane (BTESM) and BTESE or a mixture of BTESE and TEOS have been reported to form hybrid silica network with looser micro-structures (Figure 3), which facilitates gas permeation [78]. To separate hydrogen from big molecules such as MCH and TOL from the reaction mixture, BTESM- or BTESE-derived silica membranes with a slightly bigger molecular sieve can provide improved hydrogen permeability without compromising selectivity for the gas [88].

To fabricate the silica membrane, first, a porous substrate is chosen as a support. A hollow tube of α-alumina with a pore size in the range of a micron has been the preferred substrate. Since pores on the substrate wall are too big to lay a separating membrane directly on it, porous intermediate layers of materials such as γ-alumina and zirconia-silica are applied on the substrate surface by dip-coating in the respective sols followed by drying and calcining at high temperature (400–800 °C). Multiple layers are often applied by repeating this process until the pore size of the intermediate layer comes down to 45 nm and the surface is smooth and ready for final deposition of the active silica membrane by applying the silica sol on the surface followed by drying and heating at high temperature.

While CVD has advantages over sol-gel like membranes of very uniform thickness, due to the ability to control the thickness and reproducibility, the sol-gel technique does not require any special equipment to produce membranes. However, reproducibility in membrane thickness from one membrane to another is difficult to achieve, and another challenge is to apply the top layer without letting the sol solutions penetrate into intermediate layers, which would clog the pores and eventually affect hydrogen permeability of the membrane.

Oda et al. [90] and Akamatsu et al. [91] used dimethoxydiphenylsilane (DMPDS) as a precursor to fabricate silica membrane by counter diffusion CVD method on the γ-alumina layer coated on the outer surface of α-alumina tubular substrate, which housed 1% Pt/Al_2_O_3_ as catalyst for dehydrogenation reaction. Membranes showed permeances in the range of 10^−6^ mol.m^−2^ s^−1^ Pa^−1^ for H_2_, 5 × 10^−8^ mol.m^−2^ s^−1^ Pa^−1^ for N_2_ and below 10^−10^ mol.m^−2^ s^−1^ Pa^−1^ for sulfur hexafluoride (SF_6_). A high permeance of H_2_ compared to that of SF_6,_ which has a kinetic diameter (0.55 nm) very close to that of MCH (0.60 nm) or TOL (0.59 nm), ensures a high purity of hydrogen permeating through the membrane. Permeances did not change significantly over a temperature range from 373 to 573 K and membranes also showed a very good stability over a continuous testing period of a remarkable 1054 h with only a small decrease in the permeance for hydrogen. Zhang et al. studied the performance of silica membrane fabricated by the CVD method using triphenylmethoxysilane (TPMS) as silica precursor [71]. These membranes demonstrated a similar level of permeances for hydrogen and SF_6_ as those for membranes made from DMPDS, with a H_2_/hydrocarbon selectivity of 30,000 which allows for the production of high purity hydrogen. The TPMS-derived silica membrane demonstrated high hydrogen separation performance, albeit with a very small decrease, and excellent regeneration, and very high stability of at least over 120 h of testing with H_2_−MCH−TOL ternary gaseous mixtures.

Kanezashi et al. [92] fabricated membranes by the sol-gel method using three different silica precursors, namely, triethoxyfluorosilane (TEFS), a pendant-type alkoxysilane with a Si−F bond, TEOS-NH_4_F, and TEOS, and evaluated the effect that a source of fluorine and calcination temperatures exerted on the network pore size and gas permeation properties and compared their performances with membranes prepared from the commonly used TEOS precursor. A TEFS membrane calcined at 350 °C showed high H_2_ permeance (2 × 10^−6^ mol.m^−2^ s^−1^ Pa^−1^) and a high level of selectivity for H_2_ over large molecules (H_2_/SF_6_: >18,000) and showed approximately the same values for gas permeance and pore size as that of a F-SiO_2_ (TEOS−NH_4_F) membrane with F/Si monomer ratio of 2/8, despite having a higher F/Si monomer molar ratio of 1/1. The TEFS membrane showed approximately the same pore size distribution and gas permeance regardless of the calcination temperature (350 and 550 °C), whereas the TEOS membranes calcined at 550 °C showed greatly decreased permeance but improved selectivity for hydrogen. This effect was attributed to lower densification of the SiO_2_ structure in the TEFS membranes compared to the TEOS membranes.

Table 3 summarises fabrication methods and properties of silica-based membranes used for the separation of hydrogen in MCH-TOL systems reported in recent literatures. 

Few studies have been carried out on BTESE-derived membranes to study their performances as a hydrogen separation membrane from the mixture of MCH-TOL-H_2_ mixtures. Niimi et al. [79] showed that BTESE-derived silica membranes exhibited a high hydrogen permeance, with selectivity of hydrogen over TOL depending on the H_2_O/BTESE molar ratio during hydrolysis stage of BTESE. The H_2_/TOL selectivity increased from 100 to 10,000 by increasing the H_2_O/BTESE molar ratio from 6 to 240 during sol preparation, while maintaining the hydrogen permeance above 10^−6^ mol.m^−2^ s^−1^ Pa^−1^. Compared with silica membranes fabricated by CVD discussed above, these BTESE membranes derived by sol-gel have lower selectivity for TOL and SF_6_, nevertheless H_2_ produced from these membranes were of very high purity [33,79].

## 4. Catalytic Membrane Reactor

In the earlier days of MCH dehydrogenation processes, simple fixed bed type reactors were mostly utilized for the reaction followed by separation of TOL and hydrogen. To stretch the limit of MCH-TOL equilibrium conversion, researchers had repeatedly proven that continuous removal of H_2_ made it possible to exceed the equilibrium values of conversion [32,95]. This can be made possible by coupling a simple catalytic dehydrogenation reactor unit with a H_2_-selective membrane, aiding the equilibrium shift to attain a higher conversion Equation (1). Another advantage is that high-purity H_2_ can also be obtained easily without additional post-treatments [91]. Generally, two configurations of reactor systems have been proposed to incorporate the use of these thin but highly selective membrane layers in conjunction with a traditional catalytic reactor system, either via ex-situ or in-situ hydrogen separation set up. Figure 4 illustrates the differences between these configurations. Ex-situ configuration is easy, straightforward, and good for retrofitting existing technology; however, it can be expensive as more operating units are then required. In recent years, CMR combining catalytic reaction and membrane separation into a single unit (i.e., in-situ separation) have attracted research attention for MCH hydrogenation. For example, the use of palladium (Pd) or Pd-alloy membranes in CMR, where the membrane “extracts” hydrogen from a reaction, have been proved experimentally and theoretically to be efficient in enhancing conversions and/or lowering operating temperatures of endothermic, equilibrium-limited reactions such as the dehydrogenation of MCH [44,96]. The CMR has the advantages of being compact, which ultimately lowers the capital and operating costs as no intermediate equipment is required. Typical forms of CMR are tubular, plate, and hollow fiber type reactors [97].

In 1995, Ali et al. [95] reported two plug flow reactors coupled in series with interstage tubular Pd-Ag membranes. MCH conversion of over 90% was achieved at 320–400 °C under the pressure of 1–2 MPa. The importance of the membrane was proven when they reported that the conversion was 52% worse off when the reaction was conducted without the membrane. Figure 5 shows an example of the ex-situ catalytic membrane reactor system where a packed bed reactor was placed ahead of a membrane reactor. Li et al. placed their catalytic and membrane parts in series. The catalyst used was 2 wt% Pt/γ-Al_2_O_3_/α-Al_2_O_3,_ while the membrane was Ni-doped silica fabricated via a sol-gel route and argon was applied as a sweep gas [33]. Using a similar catalytic membrane reactor configuration system set up and catalyst, Meng et al. [88] reported a much higher MCH conversion at 98% under the reaction conditions of 270 °C and 0.23 MPa, coupling with an organosilica membrane fabricated from BTESE.

As for the in-situ type hydrogen separation CMR, the catalyzed dehydrogenation reaction and the extraction of hydrogen occur in an integrated device. Over the years, many integrated designs have been proposed and the two most prominent designs are: (1) a tubular membrane reactor where the pelletized catalyst is placed in the hollow center while the membrane is coated on the tubular wall, and (2) the multilayered membrane with catalytic material being impregnated inside porous support followed by incorporation of a H_2_-selective membrane. Figure 6 and Figure 7 show examples of type (1) and type (2) in-situ CMRs, respectively. Hirota et al. [32], and Akamatsu et al. [91] demonstrated the utilization of type (1) in-situ CMRs. The former used a commercial 0.5% Pt/Al_2_O_3_ while the membrane was activated carbon prepared via vapor phase synthesis from furfuryl alcohol deposited on a tubular Al_2_O_3_ support. They showed an improved MCH conversion over the theoretical equilibrium conversion versus a case without membrane, owing to the selective removal of generated H_2_. Later in 2015, by using a similar Pt/Al_2_O_3_ catalyst but CVD synthesized DMDPS-derived membrane, Akamatsu et al. [91] successfully demonstrated the longevity of such CMR performance after conducting their test for over 1000 h, with an averaged MCH conversion of ~40% and H_2_ purity of ~99%.

Japanese researchers synthesized a multilayered catalytic tubular membrane reactor for facilitating in-situ concurrent dehydrogenation and hydrogenation processes. In 2010, Oda et al. [90] conducted a simulation via a bimodal system consisting of 1% Pt/α-Al_2_O_3_ as the catalyst and a DMPDS-based separation membrane. They predicted MCH conversion of 99% at 260 °C, under the operating conditions of GHSV 67.9 h^−1^, 0.1–0.25 MPa, and a MCH flow rate of 1.31 × 10^−6^ mol/s. Without the use of sweep gas, a production of hydrogen with purity of no less than 99.95% was achievable. Later, Li et al. [93] and Niimi et al. [79] both chose a tubular porous alumina (porosity: 50%; length: 10 cm; outer diameter: 1 cm) as the support and impregnated platinum on it, the CMR fabrication was then completed by synthesis of BTESE-derived membrane via a sol-gel route. With the aid of argon sweep gas, an improved MCH conversion of 77% was attained, compared to equilibrium conversion of 60%. Hydrogen permeance through these membranes was driven by low pressure on the other side of the membrane, thus producing hydrogen of purity close to 100%, attributed to the high H_2_/TOL selectivity of 10^4^ [79].

In addition to the aforementioned CMR types, researchers also explored alternative modules and operating conditions. Kreuder et al. [98] prepared various catalysts with Pt and Pt-Sn as the active material and Al_2_O_3_ prepared by different ways and also CeO_2_-ZrO_2_ as the support materials, with catalysts coated on the wall of thin foils inside a microstructured reactor, and compared their performances. They found that 1 wt% Pt/Al_2_O_3_ showed the best performance but reported fast deactivation of catalysts by carbon formation on the catalyst surface at normal pressure (~1 bar). They showed improved catalytic performance by redesigning microstructured reactors with pockets filled with the catalyst (fixed bed type) and using Pd foil to continuously remove hydrogen [99]. A spray pulse mode reactor was suggested by Shukla et al., where the dehydrogenation reaction was conducted in a glass reactor consisting of a heated plate. They used 3 wt% Pt/Y_2_O_5_ and 3 wt% Pt/V_2_O_5_ as the catalysts at 350 °C and under atmospheric pressure. MCH conversion of 98% and H_2_ selectivity ~100% were achieved. Y_2_O_5_ was better than V_2_O_5,_ as the former resulted in higher H_2_ evolution rate [43]. More recently, Takise et al. [57,58] and Kosaka et al. [60] employed an electric field in their fixed bed reactor to achieve high MCH conversion above thermodynamic equilibrium at low temperatures of 160–170 °C. The electric field promoted MCH dehydrogenation by surface proton hopping, following an irreversible pathway even at low temperatures.

## 5. Commercial Development

Chiyoda in Japan has taken serious initiatives toward developing MCH-TOL based commercial dehydrogenation by building a pilot plant [31] that used their proprietary Pt/Al_2_O_3_ type catalyst to achieve above 99.9% TOL selectivity and 95% H_2_ yield with a hydrogen generation rate of above 1000 Nm^3^-H_2_.h^−1^.m^−3^-catalyst for over 10,000 h of stable performance (Figure 8) coupled with an excellent accompanying hydrogenation process performance, with TOL conversion: >99%, MCH selectivity: >99%, MCH yield: >99%. Chiyoda also demonstrated that by combining these hydrogenation plants with a plant that converts electricity from renewable sources such as windmills and solar panels into hydrogen by electrolysis [31], an efficient MCH based hydrogen storage system can also be developed when produced electricity is in surplus. In another successful initiative from Chiyoda together with other member companies of the Advanced Hydrogen Energy Chain Association for Technology Development (AHEAD), an international supply chain was established in which up to 210 mt/year hydrogen in the form of MCH is shipped at normal temperature and pressure from Brunei to Kawasaki, Japan, where hydrogen is stripped from the LOHC, and then TOL is shipped back to Brunei for rehydrogenation with the hydrogen produced by steam reforming processed gas from the liquified natural gas (LNG) liquefaction plant [100,101]. A recent study shows that the transport of hydrogen in the form of LOHC by ships is significantly cheaper than pipeline transport of compressed hydrogen [102]. Chiyoda has released its roadmap toward the development of hydrogen toward the development of an international hydrogen supply chain immediately, full-fledged hydrogen power generation by 2030, and full-fledged operation of manufacturing, transportation, and storage of zero carbon emission hydrogen by 2040.

Chiyoda Company also developed a compact-type dehydrogenation facility that fits into the FCV fuel station for automatic operation [31]. Such downsized plants consist of underground MCH storage tank, a dehydrogenation reactor and a H_2_ purification unit, and a H_2_ compressor that operate automatically to deliver hydrogen to vehicles. Such compact downsized LOHC based technology to produce hydrogen can be very attractive to provide fuel to large locomotives such trains. Germany’s Siemens Mobility and Helmholtz Institute Erlangen-Nuremberg for Renewable Energy (HI ERN) are putting efforts into conducting research on the development of “on-board” LOHC technology to power railway carriers [103]. While big and small vehicles powered by fuel cells that draw hydrogen from storage tank are being produced by different manufacturers, an on-board LOHC based dehydrogenation unit is a new area of research as it requires reactors with a high power density that must follow dynamic load curves very quickly and also the energy required for dehydrogenation needs to be provided in efficient and economical way [21,104].

There have been several other initiatives in the industrial implementation of the LOHC technology [17]. There is also growing interest in the development of LOHC-based dehydrogenation plants that can provide hydrogen for other processes that require energy. A recent research considered heat-integrated combination of a hydrogen-fired 7.7 MW gas turbine and a corresponding LOHC system that generated hydrogen as a model system to examine the feasibility of a turbine that used hydrogen extracted from the preheated LOHC in a dehydrogenation reactor using waste heat from the gas turbine process [105]. This, as well as another model-based study [106], suggest that LOHC dehydrogenation plants have potential for the development of a reliable source of hydrogen for processes that require a heavy supply of hydrogen as fuel.

## 6. Challenges and Prospects

Significant advances in the commercial development of a MCH/TOL system for the future of hydrogen transportation and distribution benefits from recent progress in the development of high-performance catalysts and the reactor system design. While the hydrogenation of aromatics and the dehydrogenation of cyclic hydrocarbons are mature industrial processes in oil refining, using LOHCs such as MCH/TOL system in smaller scale will require different processes for considering the safety, robustness, and fast kinetics. There is still a need for further development, among others, in the terms of reactor systems, process heat integration, and LOHC optimization. In addition, despite technical, environmental, and economic advantages, there are still a few challenges in the development of LOHC based hydrogen on a commercial scale, a major one being the large amount of energy required to maintain a high temperature for the catalytic dehydrogenation process, together with a few other issues such as the formation of coke at a high temperature and a low flash point of MCH (–3 °C), which makes it more inflammable. These issues, especially the high energy demand for the dehydrogenation process, needs more study on scientific and engineering fronts. A recent study suggests that providing the heat for dehydrogenation from fuel cells not only improves the overall efficiencies but also lowers the costs [102]. In addition to the provision of dehydrogenation heat, there is further potential for reduction in electrolysis and fuel cell investment as well as LOHC raw material prices. In a recent study that compared the costs and benefits of MCH with liquid hydrogen and ammonia as a source of hydrogen, MCH stands well, if not over-performing compared to other two. For these reasons, MCH is ready for introduction to the market as their conversion and release systems are technologically mature, and therefore, is a serious contender as a commercial source of hydrogen in future [102]. From economic and technical perspectives, toluene and its derivative dibenzyltoluene have been reported to have second highest potential, standing very close to methanol at the top among different LOHCs; the main advantage of methanol being the low raw material prices. Methanol is produced by catalytic synthesis of CO_2_ and H_2._ However, further development on reactor systems, process heat integration, and LOHC optimization is expected to reduce the cost of large-scale production of TOL and dibenzyltoluene, making them highly preferred LOHC [102].

So far, CMR combining catalyst reaction and hydrogen separation in a single system has received increasing research attention for the MCH dehydrogenation reaction. Hydrogen extraction from the reaction mixtures results in enhancing the reaction conversion, owing to the equilibrium shift by selective hydrogen removal and the simultaneous reaction of hydrogen production at relatively lower temperatures. However, CMR still faces the challenge of the difficulty of fabricating the reactor as well as equipment servicing, which can be trickier. There is also research need for developing the MCH/TOL separation membrane to remove toluene from the product and thus shifting the equilibrium to the right and recovering toluene. For MCH dehydrogenation with near total product selectivity toward toluene and hydrogen without significant level of low molecular weight side products, the use of a silica precursor that produces membranes with bigger sieving mess size permits better removals of hydrogen without compromising its purity. Further research is needed in these directions to explore silica precursors for improved hydrogen permeability and improve the reproducibility of inorganic membranes for scale up.

## Figures and Tables

**Figure 1 membranes-11-00955-f001:**
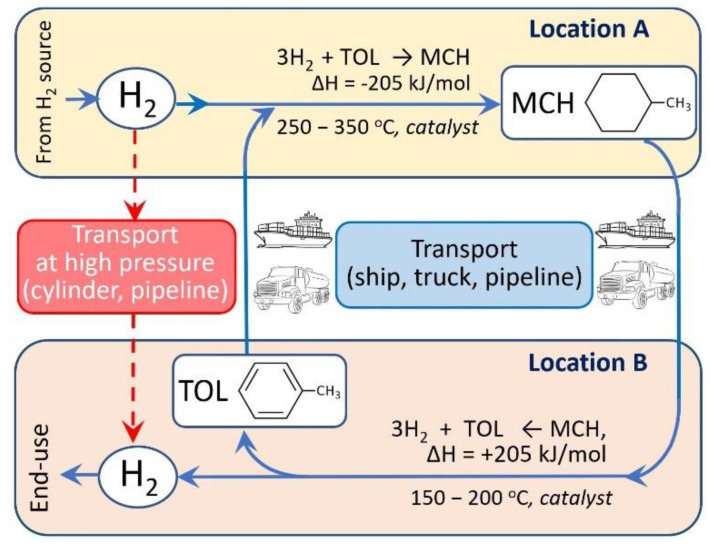
Schematic diagram of MCH-TOL based hydrogen supply chain that can utilize existing transportation infrastructures developed for gasoline, providing a much safer supply compared to the direct supply of high-pressure gaseous hydrogen.

**Figure 2 membranes-11-00955-f002:**
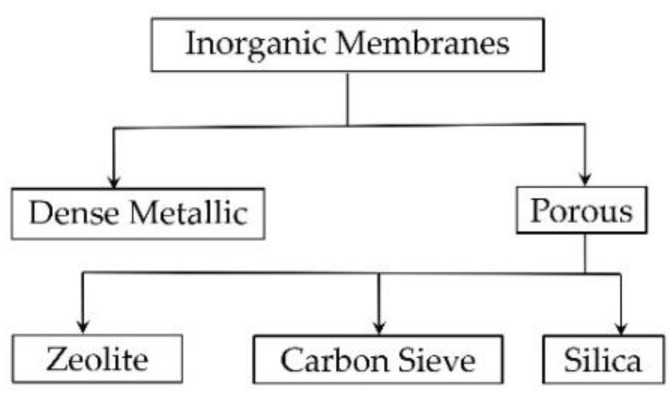
Type of inorganic membranes used for gas separation.

**Figure 3 membranes-11-00955-f003:**
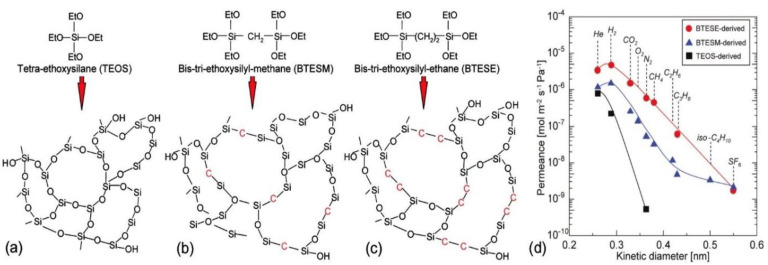
Schematic diagrams of amorphous silica networks derived by (**a**) TEOS, (**b**) BTESM, and (**c**) BTESE. The presence of (Si-CH_2_-Si) or Si-(CH_2_)_2_-Si) moieties bridging alkoxysilanes in BTESM or BTESE increases the pore size and facilitates gas permeation as shown in (**d**). Adapted from [89] with permission.

**Figure 4 membranes-11-00955-f004:**
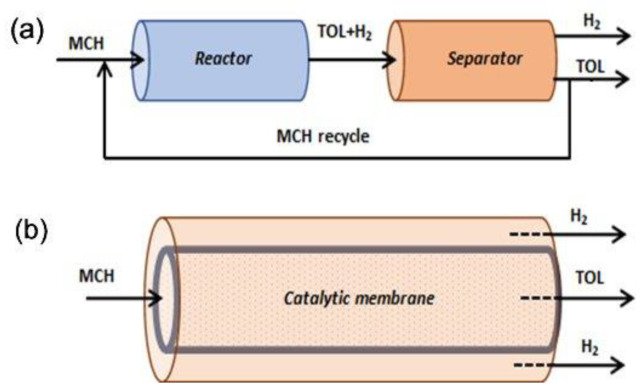
MCH dehydrogenation catalytic membrane reactor systems showing (**a**) a conventional reactor and a separator in series for ex-situ hydrogen separation (**b**) an integrated unit system for in situ hydrogen separation.

**Figure 5 membranes-11-00955-f005:**
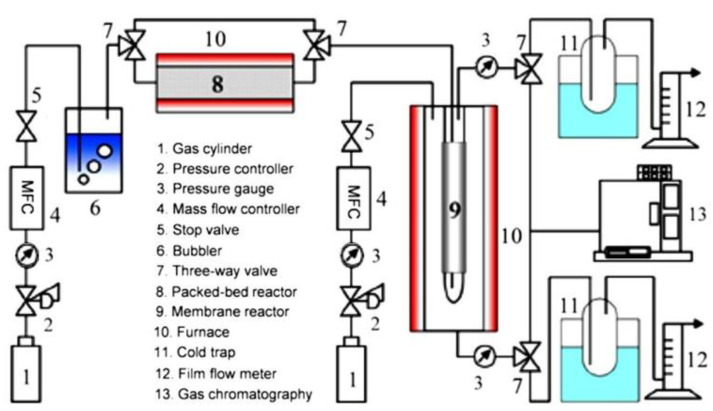
Schematic diagram of ex-situ catalytic membrane reactor system with individual numbered components shown in the included list. Adapted from [33] with permission.

**Figure 6 membranes-11-00955-f006:**
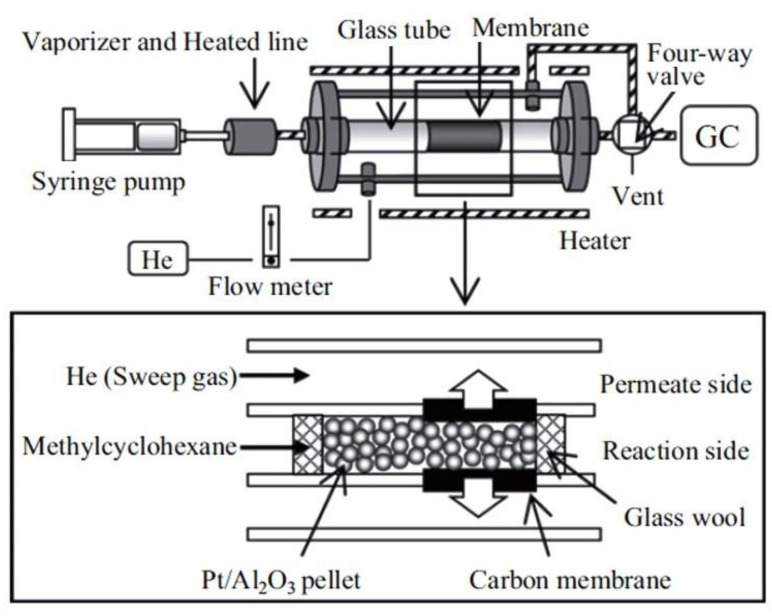
Schematic diagram of in-situ catalytic membrane reactor system: pelletized catalyst filled inside the tubular membrane reactor. Adapted from [32] with permission.

**Figure 7 membranes-11-00955-f007:**
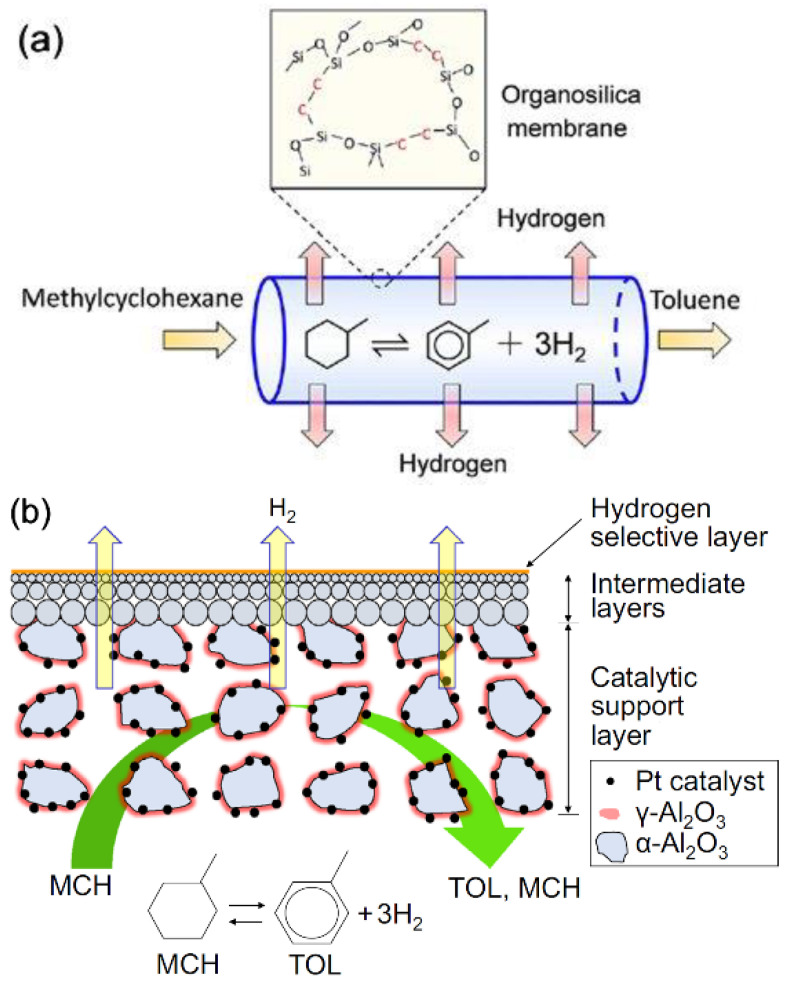
Schematic diagram of in-situ catalytic membrane reactor system. (**a**) multilayered catalytic tubular membrane reactor.; (**b**) Schematic structure of catalytic membrane. (**a**,**b**) adapted from [93] and [79], respectively, with permissions.

**Figure 8 membranes-11-00955-f008:**
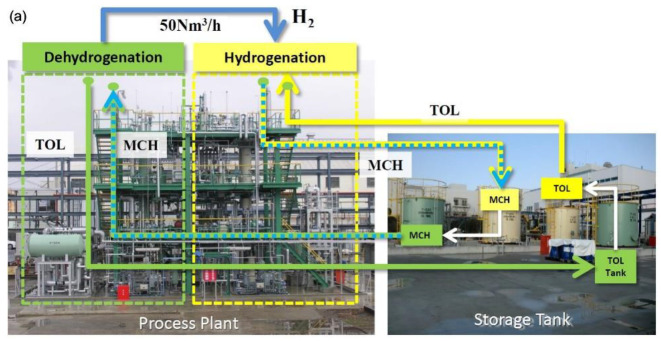
Recent initiatives taken by a consortium of some Japanese companies for transportation, storage and production of hydrogen at commercial scale based on MCH-TOL system. (**a**) Pilot scale hydrogenation and dehydrogenation plants developed by Chiyoda, Japan; (**b**) block diagram showing components of MCH-TOL based international hydrogen supply chain between Brunei and Japan; (**c**) Chiyoda’s roadmap for hydrogen supply. Adapted from [31] with permission.

**Table 1 membranes-11-00955-t001:** Physical and thermodynamic properties of some liquid organic hydrogen carriers (LOHCs) [18,30].

LOHC Pair	Dehydrogenation Enthalpy, ΔH (kJ/mol H_2_)	H-Rich, m.p./b.p. ^1^ (°C)	H-Lean, m.p./b.p. ^1^ (°C)	H_2_ Content, wt%/kgH_2_·m^−3^
MCH-TOL 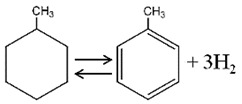	68.3	−126.6/101	−95/111	6.2/47.3
CH-BEN 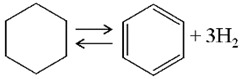	68.6	6.5/81	5.5/80	7.2/55.5
DEC-NAP 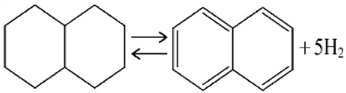	64.0 (cis) 66.7 (trans)	−43/196 (cis) −30/187 (trans)	80/218	7.3/64.9

^1^ m.p. = melting point, b.p. = boiling point.

**Table 2 membranes-11-00955-t002:** Reported catalysts for MCH dehydrogenation post 2010.

Catalyst and Support	Temp./Pressure	MCH Conversion	Reactor System ^+^	Refs., Year
0.4–1.0 wt% Pt/C	300 °C/1 bar	>95%	FB	[42], 2011
3 wt%: Pt/Y_2_O_5_, Pt/V_2_O_5_	350 °C/1 bar	98%	Spray pulse	[43], 2012
0.5 wt% Pt/Al_2_O_3_	250 °C/1 bar	60%	FB	[44], 2013
50 wt% Ni/Al_2_O_3_	500 °C/1 bar	80%	FB, HFMR	[44], 2013
10% Mo/SiO_2_	400 °C/22 bar	90% *^~^*	FB	[51], 2015
Pt, Pd, Ir, or Ni/USY zeolite	250 °C/30 bar	<10%	FB	[50], 2015
Pt/TiO_2_, Pt/γ-Al_2_O_3_	350 °C/1 bar	>80%	FB	[45], 2016
Pd/C	180 °C/1 bar	94%	FB	[49], 2016
Pt-Mn/Al_2_O_3_				
Pt/Mn/Al_2_O_3_				
Mn/Pt/Al_2_O_3_	350 °C/1 bar	90%	FB	[46], 2017
0.2 wt% Pt/Sn_x_-Mg-Al-O	350 °C/1 bar	95%	FB	[55], 2018
1 wt% Pt /Al_2_O_3_ or				
1 wt% Pt/TiO_2_ or				
1 wt% Pt/SiO_2_				
Zn, Sn, Ce, Ga, Mn added on Pt	350 °C/1 bar	>65%	FB	[47], 2018
0.4 wt% Pt/Ce_x_-Mg-Al-O	350 °C/1 bar	98.5%	FB	[56], 2019
3 wt% Pt/CeO_2_	250 °C/1 bar	51.8% ***	FB electric	[57,58], 2019
3 wt% Pt in Pt_3_Fe_0.75_Zn_0.25_	400 °C/1 bar	>95%	FB	[52], 2020
Cu-Pt /Silicalite-1 (Pt: 0.44 wt%) (Pt: 0.44 wt%)				
Cu-Pt/SiO_2_ (Pt: 1.41 wt%)	400 °C/1 bar	92%	FB	[53], 2020
2 wt% Pt/Mg-Al				
2 wt% Pt-Ir/Mg-Al	350 °C/1 bar	99.9%	FB	[59], 2020
3 wt% Pt/anatase-TiO_2_	175 °C/1 bar	37%	FB electric	[60], 2020
0.5 wt% Pt/Al_2_O_3_-TiO_2_	400 °C/1 bar	95%	FB	[48], 2020
Ga_52_Pt/SiO_2_				
Pt: 0.33 wt%, Ga: 6.1 wt%	450 °C/1 bar	16.5%	FB	[54], 2020

^+^ FB: Fix bed, HFMR: Hollow fiber membrane reactor, *^~^* Reported as TOL selectivity, *** Reported as H_2_ yield.

**Table 3 membranes-11-00955-t003:** Silica based membranes * for hydrogen separation reported in recent literatures.

Precursor	Deposition	Condition	H_2_ Permeance mol·m^−2^ s^−^^1^ Pa^−1^	Selectivity	Refs.
Dimethoxydiphenylsilane (DMDPS)	CVD	573 K	1.2 × 10^−6^	H_2_/N_2_: 3.2 × 10^1^ H_2_/SF_6_: 9.6 × 10^3 #^	[91]
DMDPS	CVD	473–573 K	6.9 × 10^−7^–1.3 × 10^−6^	H_2_/N_2_: 28–84 H_2_/SF_6_: 6900–37,000 ^#^	[90]
TPMS Triphenylmethoxysilane (TPMS)	CVD	573 K ΔP = 0.5 MPa 25 days running	~10^−6^	H_2_/hydrocarbon: 30,000	[71]
Triethoxyfluorosilane (TEFS), tetraethyl orthosilicate (TEOS), TEOS-NH_4_F	Sol coating	323–773 K	TEFS: 2 × 10^−6^ TEOS: 1.3 × 10^−6^ TEOS-NH_4_F: 2.3 × 10^−6^	H_2_/N_2_: TEFS: 8.9 TEOS: 136 TEOS-NH_4_F: 10	[92]
bis(triethoxysilyl)ethane (BTESE)	Sol-gel	473 K	(1.51−2.83) × 10^−^^6^	H_2_/SF_6_: 290–1000	[33]
BTESE	Sol-gel	473 K	8.2 × 10^−7^	H_2_/Toluene: 16,000	[79,88]
BTESE	Sol-gel	473 K	1.3 × 10^−6^	H_2_/N_2_: 34 H_2_/C_3_H_8_: 6680 H_2_/SF_6_: 48,900	[93]

* Support material is α-Al_2_O_3_ and intermediate layer is γ-Al_2_O_3_. ^#^ Estimated from gas permeance values [94].

## Data Availability

All data reported in this review paper were obtained from various scientific publications included in the Reference section.

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
