# Peer review of "Recent Advances in Catalysts and Membranes for MCH Dehydrogenation: A Mini Review"

_membranes, 2021, doi:10.3390/membranes11120955_

Round 1

Reviewer 1 Report

In the most sections of manuscript the presented contents are general and well-known information.

In the "catalysts for MCH dehydrogenation" section, general information of catalysts such as catalyst type and the value of reaction conversion are presented and no in-depth information is provided.  

In " Membranes for MCH dehydrogenation" section, various types of membranes which can be used for hydrogen separation is presented and their pros and cons are introduced which can be found in very previous published literatures. Or in the "Catalytic membrane reactor" section, membrane reactors are preliminary presented and some previously conducted work are mentioned which have no useful or new scientific tips. Most of the contents are presented in these sections are valid for all membrane processes and is not specific to the MCH dehydrogenation process.

The "Commercial development" section focused on introducing of Chiyoda corporation, without a good discussion on the challenges. 

Author Response

Response:

Authors are thankful to the reviewer for feedback.

This minireview was prepared to cover the research work done in recent years in the areas of dehydrogenation of MCH and separation and purification of hydrogen using membranes. Authors agree that many of the membranes used in MCH dehydrogenation can be used in other systems also as they share the same requirement for hydrogen separation. Authors wanted to update readers about what has been done in this area rather than choosing reactors and membranes exclusive to MCH dehydrogenation. So, the content may appear ‘general’ and applicable to many other LOHC systems rather than MCH specific due to some common feature of LOHC system. Authors believe this will not affect the scope of this minireview.  This has been recognised by the positive comments from other reviewers.

Challenges regarding Chiyoda’s initiatives discussed in “Commercial Development” section and other scaled-up dehydrogenation plants has been covered in the section “Challenges and Prospects” immediately after that. More contents have been added in the challenge section.

Reviewer 2 Report

Authors have presented the current manuscript (mini-review) well. Please check the color highlighted text and corresponding comments. I recommend the manuscript for this publication after minor corrections.   

Author Response

Response: Authors are thankful to the reviewer for providing constructive comments and pointing out errors. Responses to the reviewer’s individual comment are given below.

Line 42: Please check the typo mistakes throughout the manuscript.

Response: Manuscript has been checked and errors have been corrected.

Page 2, table 1: Please rearrange the table or move to the next page for better understanding.

Response: The table has been rearranged and the references have been moved to the bottom of the table.

Line 93: It is good to show the operating temperature range.

Response: "above 300 oC" has been added.

Line 123: It seems more good if you mention the year. Please check the earlier as well.

Response: "1966' and "1977" have been added for these works.

Table 2: Please arrange the table properly for beautification and better clarity to readers.

Response: Table has been rearranged and notes at the end of the table now appear together with the table on the same page. Authors believe pre-publication editorial work will make sure whole table remains on the same page in the final version. 

Line 204, Figure caption: Please provide few examples in each subsection.

Response: Each type of these membranes has been discussed in the body text.  

Line 321: Provide the kinetic diameter value to understand quickly.

Response: Kinetic diameters for SF6, MCH and TOL have been included.

Line 357, Caption table 3: Though other types of membranes have difficulties in terms of cost and other issues, it is better to show few important literature in the table.

Response: We appreciate the reviewer’s comment. Comprehensive reviews on membranes for hydrogen separation have been previously published elsewhere (e.g. Ref. 31 and 32).  In this work, important literatures for other membrane types used for hydrogen separation have been provided in the body text (Lines 217-219) , and only those literatures on silica based membranes used specifically for MCH dehydrogenation work have been included in the table.

Reviewer 3 Report

The abstract is to the point and well focussed - a characteristic prevalent throughout the paper.  

the arguments and selected example are presented rationally and appropriately in this review.  The level is accessible and relevant also to non-specialists with an interest in green(er) energy techniques.

Page 2:  Metal hydrides are an important class of hydrogne storage materials so perhaps some (short) elaboration of their key features, mechanism of H2 interaction and pros/cons would be appropriate here.

In Table 1 the last column heading is incoherent what is the meaning of wt%/[18]?  What is the significance of reference 18 here?  The value for MCH-Tol is 6.2/47.3 for example.  The origin of the number 47.3 is not clear to the reader.

Page 3:  Are there volatility/material loss issues that arise when using and reusing MCH-TOL?  Also a brief word on the health hazards associated with these materials would be appropriate as their handling in scaled up operations may pose a health hazard?

The concepts introduced here are clear and logical.

Line 96:  is MORE cost effective...(word missing).

Page 4:  define USY near point of first mention.

Page 13:  The commercial development section is particularly impressive and inspirational.  Reference to cyclical systems using wind turbines, solar panels/renewable sources is appropriate.

Page 15:  The honest admission here that scaling the MCH/TOL storage system up or down poses challenges.  Perhaps a statement giving the author's opinion on the greatest challenges for scaling up and for scaling down the process would provide a focal point for future research.

The challenges of the system are well summarised:  maintaining high T, coking and flash points amongst others.

Line 540 mentions methanol as a hydrogen storage contender that is even more high profile than TOL/MCH.  Why is this?  What is the mechanism of hydrogen interaction, pros/cons - a short paragraph on the main characteristics of methanol in relation to hydrogen storage would be appropriate in the introduction - given the importance referred to on line 540.

The reference selection is adequate/thorough.

The championing of chemical solutions for hydrogen storage raises the profile of our chemical discipline making this an important contribution to our science in addition to being a highly topical subject.

Author Response

The abstract is to the point and well focussed - a characteristic prevalent throughout the paper.  

the arguments and selected example are presented rationally and appropriately in this review.  The level is accessible and relevant also to non-specialists with an interest in green(er) energy techniques.

Response: Authors are very much thankful to the reviewers for constructive feedback and comments to make the minireview article even better. Author’s response to the reviewer’s comments are given below:   

Page 2:  Metal hydrides are an important class of hydrogen storage materials so perhaps some (short) elaboration of their key features, mechanism of H2 interaction and pros/cons would be appropriate here.

Response: Authors agree with the reviewer that metal hydrides, another important class of hydrogen storage system needs to be acknowledged. Following description about metal hydrides together with 4 relevant references have been added after from Line 42 of the original manuscript:

“Together with some other metal hydrides (MH) magnesium hydride (MgH2) is considered as one of the potential hydrogen storage materials because of excellent hydrogen storage capacity (7.6 wt%), low price, rich resource metals and excellent reversibility but some unfavorable properties such as high thermodynamic stability of the hydride, slow dehydrogenation and high temperature (300-400 oC) required for both dehydrogenation and hydrogenation make them unfeasible for practical use. Different approaches have been taken to address these challenges, such as by using alloys of two or more than two metals, using nanostructured metal hydrides of high porosity.”

In Table 1 the last column heading is incoherent what is the meaning of wt%/[18]?  What is the significance of reference 18 here?  The value for MCH-Tol is 6.2/47.3 for example.  The origin of the number 47.3 is not clear to the reader.

Response: Thanks the reviewer pointing out. The references are now moved to the Table 1 caption to show the sources of data to avoid any confusion . Hydrogen contents of LOHC have been expressed both in wt% (first figure) and kgH2.m-3, i.e, kg of H2 per m3 of LOHC (second figure). 

Page 3:  Are there volatility/material loss issues that arise when using and reusing MCH-TOL?  Also a brief word on the health hazards associated with these materials would be appropriate as their handling in scaled up operations may pose a health hazard?

Response: Authors agree with the reviewer’s suggestion about including a short description about the material safety. Following lines have been included in page 2 just before Table 1.  

On the other hand, high boiling temperatures of MCH and TOL (>100 oC) make them less volatile and therefore less hazardous to handle at normal temperature and pressure as suggested by a recent thermal hazard study. However, high aquatic toxicity and poor biodegradability of MCH makes it potential potentially persistent environmental pollutant and reproductive toxicity TOL suggest the LOHC system could generally present a high hazard to humans as well as the environment and make them inferior than current diesel oil based energy system.

The concepts introduced here are clear and logical.

      Response: Thanks the reviewer’s positive comment.

Line 96:  is MORE cost effective...(word missing).

Response: “More”  has been added.

Page 4:  define USY near point of first mention.

Response: “Ultra-stable Y” has been added

Page 13:  The commercial development section is particularly impressive and inspirational.  Reference to cyclical systems using wind turbines, solar panels/renewable sources is appropriate.

Response: Thanks the reviewer’s positive comment. Relevant reference has been provided.

Page 15:  The honest admission here that scaling the MCH/TOL storage system up or down poses challenges.  Perhaps a statement giving the author's opinion on the greatest challenges for scaling up and for scaling down the process would provide a focal point for future research.

Response:  One of the major challenges authors have identified is the cost effective management of heat required for dehydrogenation to maximize the efficiency reactors and even making them ‘smart, that is,” self-regulating as per the demand for hydrogen output by supplying the waste heat of fuel cells to the dehydrogenation reactor. These challenges together with other have been discussed in the last section of the manuscript.

The challenges of the system are well summarised:  maintaining high T, coking and flash points amongst others.

Line 540 mentions methanol as a hydrogen storage contender that is even more high profile than TOL/MCH.  Why is this?  What is the mechanism of hydrogen interaction, pros/cons - a short paragraph on the main characteristics of methanol in relation to hydrogen storage would be appropriate in the introduction - given the importance referred to on line 540.

Response:  One paper presented estimation of cost of storing and transporting and producing hydrogen and for different LOHCs based on economic and technical aspects where methanol system appeared marginally ahead of MCH-TOL, mainly because of the cost of production of methanol. This is rough estimation and may change with the improvements of MCH-TOL reactors and catalysts. In face there are different hydrogen storage approaches in fierce competition, and it is difficult to predict how new research affect the race. To make it clearer, relevant part of the paragraph has been revised and now it appears as below:

“From economic and technical perspectives, toluene and its derivative dibenzyltoluene have been reported to have second highest potential standing very close to methanol at the top among different LOHCs; the main advantage of methanol being the low raw material prices. Methanol is produced by catalytic synthesis of CO2 and H2. However, further development on reactor systems, process heat integration and LOHC optimization is expected to reduce the cost of large-scale production of TOL and dibenzyltoluene making them highly preferred LOHC”

The reference selection is adequate/thorough.

The championing of chemical solutions for hydrogen storage raises the profile of our chemical discipline making this an important contribution to our science in addition to being a highly topical subject.

Response:  Authors couldn’t agree more with the reviewer’s opinion as we all stand probably at the most critical time in the whole history of mankind with a big responsibility to undo the damages made to the environment by creating efficient chemical technology to produce green energy at highly affordable price.

Reviewer 4 Report

The authors of this manuscript mainly deal with the dehydrogenation process of Methylcyclohexane (MCH) to obtain toluene and H2 that over the years several studies have been carried out from the use of catalysts based on noble metals (Pt and Ir) as the use of transition metals such as: Ni, Mo, Zn, Ce and Cu, among others, achieving several advances on this particular issue, ensuring that the process is carried out at different temperature and pressure conditions, use of activated carbon until use of membranes with different materials and synthesis methods. The exhaustive research carried out by the authors is reflected in the manuscript, taking the most relevant data on the subject and the way in which this information is presented is adequate, especially the handling of the important data of each of the processes with the transcendence necessary for the reader to understand what has been done and where the research has been focused.

In addition, the authors do a whole development of the subject that includes current applications in which the process is used, being a connection of the study of this type of transformation, its benefit and its future. Therefore, I must mention that the publication must be approved.

Author Response

Response:  Authors are very much thankful to the reviewer for such a nice and positive feedback and recognizing the effort that has been put to prepare this mini review.  

Reviewer 5 Report

Durga Acharya et al. conveyed the recent advances on different aspects of catalytic dehydrogenation of MCH, including catalysts used for MCH dehydrogenation reaction, inorganic porous membranes for separation of hydrogen from reaction mixture and design of CMR for MCH-TOL system and reported some significant achievements as well on the commercial development of MCH-TOL based hydrogen storage, and transportation. The review covers a very hot topic nowadays of hydrogen storage and separation for clean and sustainable energy technology. It is informative, well-organized, simple, and English is acceptable. I recommend it for consideration for publication in membranes after a minor revision, please find my comments here:

  1. At the end of the abstract authors are advised to write one sentence summarizes their review and its importance to attract readers.
  2. I am not sure this sentence " ... their gravimetric hydrogen capacity is lower than 1% at ambient temperature and high pressure up to 100 bar" in the introduction is correct or not because recently MOFs and carbon materials are mainly used for hydrogen storage. The author should revise it very carefully.
  3. The authors should write exactly what kind of catalysts used for hydrogenation and dehydrogenation in Figure 1 (locations A and B).
  4. Figure 2 is not enough summarizing types of membranes for hydrogen separation. Recently there are tremendous research works on developing membrane for water electrolyzers and fuel cells. For example, the membranes can be divided based on charge into cationic. ionic. etc. Authors should provide more details on that. In addition, there are organic membrane (for example, https://pubs.acs.org/doi/10.1021/acsapm.1c00869), I wonder the reason the authors did not mention them in their review.
  5. Where is the conclusion?

Author Response

Response:  Authors are very much thankful to the reviewers for constructive feedback and comments to make the minireview article even better. Author’s response to the reviewer’s comments is given below:   

  1. At the end of the abstract authors are advised to write one sentence summarizes their review and its importance to attract readers.

Response:  Authors are thankful to the reviewer’s advice. Following the reviewer’s suggestion, the following summary statement has been added at the end of the abstract:

“This review mainly focuses on recent advances in different technical aspects of catalytic dehydrogenation of MCH and some significant achievements in commercial development of MCH-TOL based hydrogen storage, transportation and supply systems, along with the challenges and future prospects.” 

  1. I am not sure this sentence " ...  their gravimetric hydrogen capacity is lower than 1% at ambient temperature and high pressure up to 100 bar" in the introduction is correct or not because recently MOFs and carbon materials are mainly used for hydrogen storage. The author should revise it very carefully.

Response:  Authors thank reviewer for pointing out at this statement where further clarification is needed.

For the year 2020 US Department of Energy (DoE) has set the gravimetric and volumetric performance target for usable hydrogen storage capacity at 4.5 wt% and 30 g/L, respectively, at 233–358 K and 5–12 bars, with the ultimate target set for 6.5 wt% and 50 g/L. Some of the MOFs achieve gravimetric hydrogen capacity on par to or even better than the DoE standard but at much higher pressure and at cryogenic temperature (77 K), not at ambient temperature. At the preferred pressure range (5-12 bars), hydrogen storage capacities of MOFs drop sharply even at 77K. The line in the manuscript has been rewritten to accommodate additional information and now the complete sentence appears as:

“Other approaches for hydrogen storage are its absorption in a high surface carbon based material such as nanotubes, nanorods, graphite and activated carbon (Ref), and other non-carbon materials such as mesoporous silica (MCM-41) [Refs], high entropy alloys [Refs], metallic organic frameworks (MOFs) [Refs] have emerged but the main drawback is that their gravimetric hydrogen capacity is lower than 1% at ambient temperature and high pressure up to 100 bar, which is far below the respective gravimetric and volumetric performance target of 4.5 wt% and 30 g/L and the ultimate target of 6.5 wt% and 50 g/L at 233–358 K and 5–12 bars set by the US Department of Energy (DoE) for usable hydrogen storage capacity of on-board or mobile storage systems for the year 2020. Some of the MOFs achieve gravimetric hydrogen capacity on par to or even better than the DoE’s target at much higher pressures and at cryogenic temperature (77 K) but as the temperature is increased to as high as 160 K the capacity sharply drops.” 

  1. The authors should write exactly what kind of catalysts used for hydrogenation and dehydrogenation in Figure 1 (locations A and B).

Response:  Authors appreciate reviewer’s suggestion, but the emphasis of this figure is to illustrate how MCH-TOL based (and many other LOHC systems as well) hydrogen storage and transportation system operates, instead of specifically mentioning particular set of catalysts for hydrogenation and dehydrogenation reactions. As discussed in Section 2, there are various catalysts that can be used for MCH-TOL system.

  1. Figure 2 is not enough summarizing types of membranes for hydrogen separation. Recently there are tremendous research works on developing membrane for water electrolyzers and fuel cells. For example, the membranes can be divided based on charge into cationic. ionic. etc. Authors should provide more details on that. In addition, there are organic membrane (for example, https://pubs.acs.org/doi/10.1021/acsapm.1c00869), I wonder the reason the authors did not mention them in their review.

Response:  Hydrogen separation membranes discussed section 3 are primarily those which can withstand high temperatures required for dehydrogenation reaction of MCH (and other many LOHC). Hence, polymeric membranes which show excellent separation performance but have working temperature range below 100 oC have not been included but only inorganic membranes in Figure 2. For similar reason, membranes used in electrolyzers and fuel cells, which are mostly either polymeric or polyelectrolyte-based ion exchange (https://doi.org/10.3390/membranes11110810), and hence, were not included in Figure 2 or discussed in Section 3.

Authors believe that the modified sentence in the first paragraph of section 3 will clarify to readers why other membranes are not included.

Polymeric (organic) membranes are very common and cost-effective, however, their relatively low operating temperature range (commonly below 100 °C with some polymers below 200 °C) makes them unsuitable as hydrogen separation membrane at high temperature environment required for MCH dehydrogenation. Therefore, only membranes made of inorganic materials that can withstand high temperatures for practical applications are included in this review”.             

  1. Where is the conclusion?

Response:  Instead of giving a separate section for ‘Conclusion’, much broader and inclusive section ‘Challenges and Prospects’ has been put at the end which draws on progresses so far in the areas of catalysts, catalytic reactors and membranes, identifies challenges and points out future direction of research and development.